# Big Data in metagenomics: Apache Spark vs MPI

**José M. Abuín**[1,2]*, **Nuno Lopes**[1], **Luís Ferreira**[1], **Tomás F. Pena**[2], **Bertil Schmidt**[3]

**1** 2Ai—School of Technology, IPCA, Barcelos, Portugal, **2** CiTIUS, Universidade de Santiago de Compostela, Santiago de Compostela, Spain, **3** Department of Computer Science, Johannes Gutenberg University, Mainz, Germany

* josemanuel.abuin@usc.es

**Data Availability Statement:** All relevant data are within the manuscript and on GitHub (https://github.com/jmabuin/metacache-mpi).

**Funding:** This work has been funded by Xunta de Galicia under grant ED481B 2018/013 which was

## Abstract

The progress of next-generation sequencing has lead to the availability of massive data sets used by a wide range of applications in biology and medicine. This has sparked significant interest in using modern Big Data technologies to process this large amount of information in distributed memory clusters of commodity hardware. Several approaches based on solutions such as Apache Hadoop or Apache Spark, have been proposed. These solutions allow developers to focus on the problem while the need to deal with low level details, such as data distribution schemes or communication patterns among processing nodes, can be ignored. However, performance and scalability are also of high importance when dealing with increasing problems sizes, making in this way the usage of High Performance Computing (HPC) technologies such as the message passing interface (MPI) a promising alternative. Recently, MetaCacheSpark, an Apache Spark based software for detection and quantification of species composition in food samples has been proposed. This tool can be used to analyze high throughput sequencing data sets of metagenomic DNA and allows for dealing with large-scale collections of complex eukaryotic and bacterial reference genome. In this work, we propose MetaCache-MPI, a fast and memory efficient solution for computing clusters which is based on MPI instead of Apache Spark. In order to evaluate its performance a comparison is performed between the original single CPU version of MetaCache, the Spark version and the MPI version we are introducing. Results show that for 32 processes, MetaCache-MPI is 1.65× faster while consuming 48.12% of the RAM memory used by Spark for building a metagenomics database. For querying this database, also with 32 processes, the MPI version is 3.11× faster, while using 55.56% of the memory used by Spark. We conclude that the new MetaCache-MPI version is faster in both building and querying the database and uses less RAM memory, when compared with MetaCacheSpark, while keeping the accuracy of the original implementation.

## Introduction

Continuous advances in next generation sequencing (NGS) technologies have led to a constant production of huge amounts of genomic data. These big quantities of data need to be analyzed

awarded to José M. Abuin. This project was also funded by national funds (PIDDAC), through the FCT - Fundação para a Ciência e Tecnologia and FCT/MCTES under the scope of the project UIDB/05549/2020 and UIDP/05549/2020 and by the project "NORTE-01-0145-FEDER-000045", supported by Northern Portugal Regional Operational Programme (Norte2020), under the Portugal 2020 Partnership Agreement, through the European Regional Development Fund (FEDER). The funders had no role in study design, data collection and analysis, decision to publish, or preparation of the manuscript.

**Competing interests:** The authors have declared that no competing interests exist.

and interpreted by domain scientists in order to obtain research results or provide proper diagnosis to patients. However, corresponding processing times can vary from several hours to even days. In addition, more detailed analysis may not be carried out because of large memory requirements that often exceed the capacity of individual workstations or even shared memory multiprocessors.

Selecting a suitable technology to deal with a critical resource (CPU time, amount of memory, etc.) can be a decisive factor that has significant influence on performance in large-scale NGS processing pipelines. Traditionally, these problems have been addressed by using HPC and low level programming languages, such as C, C++ or Fortran [1–3]. Recently, Big Data technologies such as Apache Hadoop [4] and Apache Spark [5, 6] are being employed. They allow the usage of high-level programming languages, such as Java, Python, or Scala, while providing ease of use and performance [7–11].

For many areas, the decision of which technology to use is not clear. Apache Spark offers an easy-to-use programming interface that allows the programmer to deal with large amounts of data in a quick, parallel, and easy way. These characteristics can be useful when dealing with scientific problems that need to perform simple operations on huge volumes of data. On the other hand, for High Performance Computing (HPC) environments, other alternatives, such as MPI (Message-Passing Interface) [12], can be more appropriate [13–16]. When the operations to be performed on the data are complex and require more time and memory resources, the use of Big Data technologies may not provide the desired performance. This last statement is investigated in this work by conducting a comparison between Spark and MPI using metagenomics as a case study.

Metagenomics can be widely defined as the study of genetic material gathered directly from environmental samples. In this work, the focus is set on the analysis of NGS data obtained from high-throughput shotgun sequencing of DNA samples from foodstuff, called All-Food-Sequencing (AFS) [17, 18]. AFS allows for the detection and quantification of food ingredients and microbiota. In comparison to traditional techniques such as quantitative real-time polymerase chain reaction (qPCR) [19], AFS has the ability to screen for a wide range of species as it does not require any prior definition of possible target species.

However, the original AFS pipeline [17, 18] relies on a read alignment tool (such as BWA [20–22], Bowtie2 [23], or CUSHAW [3]) for each considered reference genome. Thus, runtime increases linearly with the number of considered species, which makes this approach unsuitable for broad-scale screening with large amounts of reference genomes from various kingdoms of life.

Recent publications [24, 25], show that a $k$-mer-based exact matching approach can achieve high classification accuracy while being orders-of-magnitude faster than the alignment-based AFS pipeline. It relies on building a database of substrings of length $k$ of each considered reference genome. A sequencing read is then classified by querying the database using its $k$-mers as queries. If a query returns a match, corresponding counters for the matching reference genomes are incremented. Finally, a read is taxonomically labeled based on high-scoring counters and coverage analysis.

While similar approaches have worked well for metagenomic analysis of bacterial genomes (e.g. [26, 27]), the significantly higher complexities of eukaryotic reference genomes, relevant for monitoring food ingredients, makes the adaption of this method to food-monitoring challenging. The software that comes with these publications is *MetaCache*. It employs an intelligent subsampling of $k$-mers based on minhashing in order to reduce both memory consumption and database construction times. Nevertheless, when considering large-scale reference genome collections, this solution is still limited by the available memory on a single workstation. To deal with this limitation, it is possible to split the database into multiple parts,

consequently lowering the classification speed. Alternatively, it is possible to use the Apache Spark based version of this approach, named MetaCacheSpark, as presented in [25].

The huge amount of memory consumed by this approach, together with its high computation time, are the elements that make this case an ideal candidate for this study. Starting from this approach, an MPI variant has been developed and a comparison with both the Spark variant and the original MetaCache implementation (using a sequential approach) has been made.

Our results show that memory consumption depends on the utilized technology and on the programming language [28]. Furthermore, the speed is quite different between technologies. Experiments running MPI show an improved memory usage, some times being reduced down to 50%, while, at the same time, being faster by a factor of 2× or 3×, in all the phases involved in the AFS metagenomic analysis.

## Background

Scientific applications can be roughly divided into two groups: those that use complex models applied over a small set of data and those that have to analyze a large amount of information. The former often emerges in fields such as Engineering, Physics, or Climatology, while the latter are typical of areas like Bioinformatics, Remote sensing, Sociology, or Management. Modelling a complex system requires the use of High-Performance Computing (HPC) infrastructures, made up of powerful high-end machines with high-bandwidth low-latency networks (like Infiniband or Aries interconnect) and complemented with coprocessors such as GPUs or FPGAs. On the other hand, analysis of very large amounts of data can be performed using Big Data technologies, that are typically built on top of "commodity" hardware: large clusters of low-end machines (or virtual machines in a cloud) linked together using high-latency traditional LAN network (mostly Ethernet based).

Nowadays, the HPC community is involved in a race between companies, institutions, and research centres to reach the exascale milestone. Exascale computing refers to supercomputers capable of executing $10^{18}$ floating point operations per second (FLOPs), i.e., one exaFLOP per second. To reach this performance, future supercomputers require data delivery to be fast and efficient, both from memory and disk, and also across the network and between processors. This is a difficult task to achieve in big supercomputers, and also in large computations, like those present in scientific and data analytics problems. Also, developers will need exascale Application Programming Interfaces (APIs) to facilitate the exploitation of exceptional amounts of parallelism in applications, to enable the processing of significant amounts of data, and to support different architectures, including those based upon heterogeneous cores or accelerators. Those APIs and their implementations will need to carefully manage different kinds of memories within each node. Moreover, the need to conserve energy has led to an increased focus on reducing data motion at all levels of the memory hierarchy, from low cache levels to main memory, requiring a rethinking of algorithms as well as of the entire HPC software stack. In addition, exascale execution software systems will need to ensure that jobs continue to run despite the occurrence of system failures and other kinds of hardware or software errors.

Big Data technologies, on the other hand, have become increasingly popular, and their usage is not longer restricted to data analytics, but has been successfully used in fields like bioinformatics [7–11, 15], chemistry [29, 30], or medicine [31, 32]. Technologies like Apache Hadoop [4] or Apache Spark [5] offer a scalable way to process enormous amounts of data in large clusters of "cheap" computers or virtual machines in the cloud, using simple programming models. To improve scalability in systems with high latency networks, these technologies bring computing to data, making use of distributed file systems like HDFS [33] and trying to

process in each node only the information that lives close to it. Besides, to cope with the frequent failures of low-end computers, these technologies implement fault-tolerant mechanisms that guarantee that the jobs will continue running in the presence of hardware or software failures. Finally, Big Data processing technologies use parallel programming paradigms (such as MapReduce or graph execution models) that allow programmers to focus on the problem at hand, without having to deal with explicit I/O or data exchange among processors.

## MPI vs Spark

HPC systems and Big Data platforms rely on different programming paradigms [34]. For HPC systems, the Message Passing Interface (MPI) [12] remains the standard for large-scale parallel computing. MPI defines an interface that specifies the synchronization and data exchange of messages across nodes within a cluster. MPI function calls allow the programmer to specify the message contents to be exchanged. The functions are explicitly stated and are implemented through libraries that handle the inter-node communication.

In the case of Big Data, Apache Spark is one of the most popular technologies. Spark is based on a graph execution model, in which computation is represented by a Directed Acyclic Graph (DAG) of tasks. From the source code, Spark creates a *logical plan*, which is a DAG representing the operations to be carried out. Then, this logical plan is optimized and a *physical plan*, which specifies how the logical plan will execute on the cluster, is created [35].

Spark expresses parallelism by means of data structures, namely DataFrames, DataSets and RDDs (Resilient Distributed Dataset) that either are automatically partitioned and distributed across the cluster nodes or have been previously stored in a distributed file system, for example, HDFS. A driver process is in charge of distributing work across the executors, which are daemons running at each node and that manage the local execution of operations on local data. Spark relies on a cluster manager (usually YARN [36] or Mesos [37]) to control physical machines and allocate resources to applications.

Both technologies were designed for distributed memory systems, composed of multiple nodes with local independent memory. In terms of the programming model, Spark follows a global shared memory design, where DataFrames, DataSets or RDDs are used as global variables over a sequential program. All the data distribution and message transfer is under the control of the Spark engine. On the other hand, MPI usually follows a SPMD (Single Program, Multiple Data) model, with multiple autonomous processors simultaneously executing the same program, sending and receiving messages to and from each other, all under the programmer control. It is the developer's responsibility to distribute the data and to design the message exchange calls in such a way that the execution of the same program on all nodes achieves the desired outcome.

Thus, programming with MPI requires the developer to take explicit control of the individual node states and communication patterns. On the other hand, Spark's design is simpler for the developer, since it just needs to specify a sequential program behaviour that automatically generates local execution tasks to be distributed across nodes by the engine itself. Another added value from using Spark is the fault tolerance feature of the data and computation, i.e., if a node fails, Spark will recover the data and recompute as necessary, whereas fault tolerance must be explicitly built on MPI programs.

A main drawback of Apache Spark is that its programming model may not be suitable for all problems, e.g. irregular ones. Programmers have to express their algorithms as a set of transformations and actions over Spark data structures, having limited control over data partitioning and computing distribution. MPI, instead, provides much greater flexibility, being suitable for most algorithms, at the cost of a greater programming effort.

## Implementation

In this section details about the implementation of MetaCache-MPI will be presented. Two of the functionalities from the original MetaCache algorithm have been developed in MPI: database construction/building and database querying/classification. The main goal is to reduce memory consumption and computing times for both of these phases.

### Parallel building phase

The original MetaCache approach is to build a huge hashmap by applying a technique named *minhashing*, as described in [24]. In order to build the database from a set of input reference genomes it follows these steps:

1. Read and build the given input taxonomy.

2. Add the input sequences to the database (hashmap). Meanwhile, look up for the parent sequence and update the taxonomy.

3. Rank targets that remain unranked in the database by using external post-process files.

4. Remove overpopulated features from the database (optional).

5. Write database to disk.

MetaCache-MPI performs Step 2 in parallel by splitting the number of sequences among the MPI processes which reduces the amount of memory needed per node. At the same time, it will reduce the computing time needed to build the database. To do this, each process reads the input sequences from the input files as in the sequential case, as the sequences are needed to build the full taxonomy. However, only the sequences that correspond to the current MPI process will be added into the local hashmap. In this way, each process will contain a portion of the huge hashmap from the original case.

In order to determine whether a sequence belongs to an MPI process or not, a Round Robin algorithm is used. According to this algorithm, sequence read number $s$ will be added to MPI process number $s$ mod $N$, being $N$ the total number of MPI processes. In this way, sequence 0 will be added into process 0, number 1 into process 1, and so on. Sequence $N$ will be added again in process 0, and this Round Robin algorithm continues until there are no more sequences left.

Steps 3 to 5 are run in a sequential way in all the MPI processes, except for step number 4, which requires a reduction phase. This reduction is needed because the number of items that belong to each key has to be known. If certain key goes beyond a given number of items, it can be deleted in all the MPI processes local hashmaps. This fact is related to the *remove overpopulated features*, explained in later sections.

In order to perform this kind of reduction operations MPI provides some useful functions. However, in this case, it is not possible to use them. For example, to use *MPI_Reduce*, the same number of items must be present in each one of the MPI processes, and that is not the case. A workaround can be a combination of *MPI_Gather* and *MPI_Bcast* to gather all items into MPI process number 0, do the reduction and send the result to all the other processes. However, one of the parameters of *MPI_Gather* is an integer that represents the number of items, and in this case, the number of items can be larger than the maximum value of the integer type.

Because of these limitations, the use of an alternative is required. There are some known solutions for this problem, for example, to define specific data types, but this is not suitable for this case, as it is not possible to know the specific size that this new data type is going to need.

## MPI processes

**Fig 1. Reduction algorithm example.**

To overcome this problem our approach is to reduce the number of items per hashmap key in one of the MPI processes (Process 0) and then broadcast to all processes the list of keys that must be removed, but without using *MPI_Gather*. To do this, it is necessary to perform specific send/receive calls in the processes to get the needed data in the MPI Process 0. This is a slow process, as each one of the MPI processes is sending its data to process 0 in a sequential way. Thus, we have implemented a custom reduction algorithm to deal with this situation.

This customized algorithm requires each MPI process to be classified as a sender or a receiver. This behaviour is illustrated in Fig 1 using an example. In the first iteration of this example, Process 1 sends data to Process 0, Process 3 to Process 2, and so on. In the next iteration Process 2 becomes a sender, and sends data to Process 0, meanwhile Process 4 (which is now a sender) stays without performing any action, as it still does not have a receiver to send its data to. In the third and final iteration, Process 4 sends its data to Process 0, finalizing the reduction process. At the end, all the data is stored at Process 0, which was this algorithm final goal. If a traditional send/receive algorithm is used here, the total number of iterations would be 5, this is, the total number of processes minus one.

### Parallel classification phase

The distributed hashmap from the building phase is stored in a number of files equal to the number of MPI processes being used. At the start of the classification phase each process loads one of the database files to main memory and, in this way, reads the partial hashmap and the full taxonomy.

After that, the steps involved in the classification are:

1. Once the partial hashmaps have been read by the distributed processes, all the processes start to read the input sequences to be classified. It is important to notice here that all the processes are going to read all the input sequences and store partial results from their piece of the distributed hashmap. At this phase, each one of the MPI processes can use various threads, where each thread handles a given number of input sequences. Each thread accesses the hashmap and stores the possible candidates in the same way as in the original version.

2. Results from the previous phase are partial, and they need to be grouped. So, another reduction is needed in order to obtain the global results. To do this, the reduction algorithm that has been used in the construction phase is performed again, but now using the data of the possible candidates instead of the data used to get the overpopulated features during building. It is important to notice that this reduction phase is not multi-threaded, as there are possible deadlocks in the process.

3. After the reduction process, all the global results are going to be stored in the Process 0, where the final global candidates are obtained in the same way than the original MetaCache tool. In this final process, the same number of threads than in Phase 1 are used.

4. Process 0 writes to disk the obtained results.

5. If more sequences are left, the algorithm goes back to Step 1.

To better explain this process, a small example is shown in Fig 2, where the X in Step 1 is the first sequence to classify in the current iteration. In this example, two MPI processes are used to perform this phase, Process 0 and Process 1. Each one of these two processes is using 4 threads. At Step 1 each thread processes $S$ sequences, but all the MPI processes are going to take as input the same sequences. As 4 threads are being used, the total amount of sequences processed is $4 \times S$. It is important to notice here that this $S$ parameter can be configured by the user, and it should be set according the total number of sequences. In Step 2 the partial results are obtained, and the reduction process is performed. After that, in Step 3, four threads are used again to get the best candidates from the reduction phase. Again, each thread is in charge of process results for $S$ sequences. In the next step, MPI Process 0 is responsible of writing the final results, as this is the only process that has all the results. Finally, if more sequences are left, the algorithm goes back to Step 1.

## Results

In this section the results about performance and quantification of MetaCache-MPI are presented, including a comparison with MetaCacheSpark. In order to provide a fair comparison between MPI and Spark solutions, the same hardware infrastructure is employed. In this case, we use a Big Data cluster composed of 12 Dell EMC PowerEdge R730 servers, each one with a dual Xeon E5-2630v4 (2.2GHz 10 cores) CPUs with 384 GB RAM and 32 TB HDDs, running Java version Openjdk 1.8.0_201, gcc 7.3.1, Spark 2.2.0, and Hadoop 2.7.3. Regarding MPI, the used version was OpenMPI 3.0.2. The cluster network is a 10 GB BaseT ethernet. This infrastructure may not the best choice in terms of MPI performance, but, as was stated at the beginning of this paragraph, the main goal is to carry out a fair comparison.

Furthermore, in order to analyze the scalability of the MPI version we use a HPC infrastructure, the cluster FinisTerrae II available at the Galician Supercomputing Center (CESGA, www.cesga.es). This cluster is composed by 306 nodes, each with 2 Haswell 2680v3 processors (24 cores) and 128 GB of RAM memory. These nodes are connected with an Infiniband FDR@56Gbps network. Software versions are gcc 6.4.0 and OpenMPI 2.1.6. This cluster also

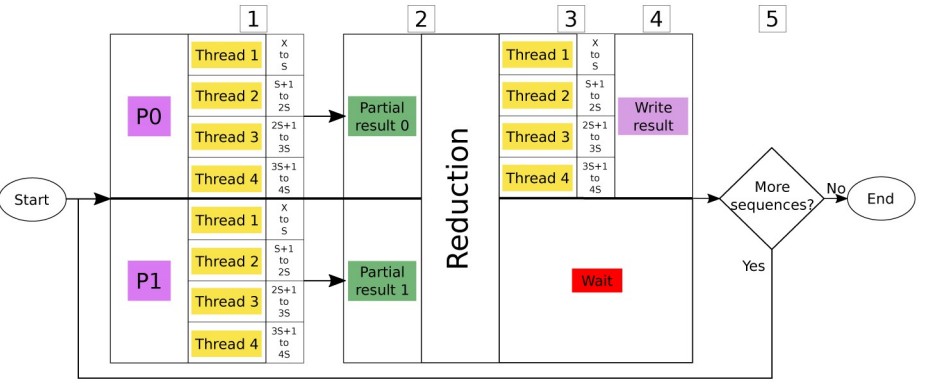

**Fig 2. Classification example.**

has other kind of nodes with accelerators, but for these experiments only the indicated type of nodes have been used.

## Parameters used during executions

The following parameters have been used during the experiments, as they are the same as used in [25]. For building:

- **-remove-overpopulated-features**: Removes from the database features that have more than a given number of items (default is 254). This parameter was introduced in previous sections, and it is explained here. For animal genomes, it happens quite often that a given key from the database is filled with locations from only a few or just a single animal, the ones that are inserted first when building the database. This leads to a huge bias in the database. For this reason, these full key values are discarded, in order to reduce the bias.

  For classification:

- **-abundance-per species**: Groups final results per species and perform a re-distribution of items classified above species level.

- **-lowest species**: The lowest common ancestor used for classification is at species level.

- **-maxcand 4**: The maximum number of candidates considered for classification is 4.

- **-hitmin 4**: The minimum number of hits to consider a candidate is 4.

- **-hitdiff 80**: Parameter used when classifying in order to determine possible candidates.

## Input data

In order to measure performance and accuracy of the MetaCache-MPI, several databases of varying size containing different organisms have been created. Food-related genomes (selection of main ingredients) used for database construction are listed in Table 1 while the considered bacteria, viruses, and archaea from NCBI RefSeq (Release 90) are summarized in Table 2. The created databases with their included reference genomes are described in Table 3.

Ten short read data sets are used for testing the performance of the classification phase. They were sequenced from calibrator sausage samples and were downloaded from ENA project ID PRJNA271645 (Kal_D and KAL_D) and PRJEB34001 (all other data)). These ten sausages data sets contain admixtures of a set of food relevant ingredients (chicken, turkey, pork, beef, horse, sheep) sequenced on an Illumina HiSeq machine as described in [38]. Table 4 shows the read data sets together with the corresponding percentage of meat components used during preparation. The samples comprise meat proportions ranging from 0.5% to 80% and can be subdivided into two categories: Kal A-E consist only of mammalian meat, while KLyo A-D represent Lyoner-like sausages containing poultry in addition to mammals [19, 39]. The data set KAL_D is identical to Kal_D but sequenced with higher coverage.

## MPI vs Spark performance results

For the comparison between the MPI and Spark approaches, two important measures should be defined. First, the time spent for both database construction and database classification. Second, the amount of RAM memory used in both phases. For the memory measurement, the maximum amount (peak) in one of the processes is used as reference.

**Table 1. Food-related reference genomes used for database construction.**

| Item | Name | ID | # sequences | Size on disk |
|---|---|---|---:|---:|
| 1 | Sus scrofa (pig) | GCA_000003025.6 | 612 | 2.4 GB |
| 2 | Equus caballus (horse) | GCF_000002305.2 | 9 636 | 2.4 GB |
| 3 | Meleagris gallopavo (turkey) | GCF_000146605.2 | 231 286 | 1.2 GB |
| 4 | Mus musculus (house mouse) | GCF_000001635.25 | 239 | 2.7 GB |
| 5 | Gallus gallus (chicken) | GCF_000002315.4 | 464 | 1.1 GB |
| 6 | Ovis aries (sheep) | GCF_000298735.2 | 5 466 | 2.5 GB |
| 7 | Rattus norvegicus (Norway rat) | GCF_000001895.5 | 955 | 2.8 GB |
| 8 | Bos taurus (cattle) | GCF_000003055.6 | 3 143 | 2.6 GB |
| 9 | Bubalus bubalis (water buffalo) | GCF_000471725.1 | 366 982 | 2.8 GB |
| 10 | Oryctolagus cuniculus (rabbit) | GCF_000003625.3 | 3 241 | 2.6 GB |
| 11 | Capreolus capreolus (Western roe deer) | GCA_000751575.1 | 3 088 511 | 3.0 GB |
| 12 | Struthio camelus australis (African ostrich) | GCA_000698965.1 | 6 914 | 1.2 GB |
| 13 | Anas platyrhynchos (mallard) | GCF_000355885.1 | 78 488 | 1.1 GB |
| 14 | Capra hircus (goat) | GCF_001704415.1 | 29 907 | 2.8 GB |
| 15 | Cervus elaphus hippelaphus (red deer) | GCA_002197005.1 | 11 479 | 3.3 GB |
| 16 | Cavia aperea (Brazilian guinea pig) | GCA_000688575.1 | 3 131 | 2.6 GB |
| 17 | Camelus ferus (Wild Bactrian camel) | GCF_000311805.1 | 13 334 | 1.9 GB |
| 18 | Canis lupus familiaris (dog) | GCF_000002285.3 | 3 268 | 2.3 GB |
| 19 | Felis catus (domestic cat) | GCF_000181335.3 | 4 508 | 2.4 GB |
| 20 | Homo sapiens (human) | GCF_000001405.38 | 594 | 3.1 GB |
| 21 | Equus asinus asinus (ass) | GCA_003033725.1 | 9 021 | 2.2 GB |
| 22 | Rangifer tarandus (reindeer) | GCA_004026565.1 | 1 360 739 | 2.9 GB |
| 23 | Phasianus colchicus (Ring-necked pheasant) | GCA_004143745.1 | 39 677 | 987 MB |
| 24 | Glycine max (soybean) | GCF_000004515.5 | 1 192 | 946 MB |
| 25 | Zea mays (maize) | GCF_000005005.2 | 267 | 2.1 GB |
| 26 | Triticum aestivum (bread wheat) | GCA_900519105.1 | 22 | 14.0 GB |
| 27 | Secale cereale (rye) | GCA_900002355.1 | 1 581 707 | 1.8 GB |
| 28 | Hordeum vulgare (barley) | GCA_004114815.1 | 1 856 | 3.8 GB |
| 29 | Oryza sativa Japonica Group (Japanese rice) | GCF_001433935.1 | 58 | 362 MB |
| 30 | Arachis hypogaea (peanut) | GCF_003086295.1 | 21 | 2.4 GB |
| 31 | Saccharomyces cerevisiae S288C (baker's yeast) | GCA_000146045.2 | 16 | 12 MB |
| **Total** | | | **6856734** | **77 GB** |

To perform these experiments, the biggest of the data sets has been used for building (AFS31RS90), and also the biggest of the sequenced sausages data has been used for classification (KAL_D).

Results about the execution time for building the database are presented in Fig 3. Here, results with one process are obtained by using MetaCache [25]. In this figure we can see how

**Table 2. Reference genomes from NCBI RefSeq (Release 90) used for database construction.**

| Organism | Number of references | Size on disk |
|---|:---:|:---:|
| bacteria | 21365 | 41.0 GB |
| viral | 10069 | 269 MB |
| archaea | 406 | 656 MB |
| **Total** | **31840** | **41.9 GB** |

**Table 3. Data sets used for database construction.**

| Name | Number of references | Size on disk |
|------|---------------------|--------------|
| **AFS20** | Animal genomes from 1 to 20 | 45.8 GB |
| **AFS20RS90** | Animal genomes from 1 to 20 plus NCBI RefSeq (Release 90) | 87.5 GB |
| **AFS31** | Animal genomes from 1 to 31 | 76.8 GB |
| **AFS31RS90** | Animal genomes from 1 to 31 plus NCBI RefSeq (Release 90) | 118.5 GB |

**Table 4. Calibrator sausage datasets and their meat composition.**

| Name | #Reads (paired-end) | Size | Cattle | Sheep | Pig | Horse | Chicken | Turkey |
|------|---------------------|------|--------|-------|-----|-------|---------|--------|
| KLyo_A | 401K | 241 MB | 14.0% | 0.0% | 80.0% | 0.0% | 0.5% | 5.5% |
| KLyo_B | 302K | 175 MB | 36.0% | 0.0% | 58.0% | 0.0% | 2.0% | 4.0% |
| KLyo_C | 507K | 298 MB | 58.0% | 0.0% | 36.0% | 0.0% | 4.0% | 2.0% |
| KLyo_D | 417K | 238 MB | 80.0% | 0.0% | 14.0% | 0.0% | 5.5% | 0.5% |
| Kal_A | 830K | 494 MB | 1.0% | 9.0% | 35.0% | 55.0% | 0.0% | 0.0% |
| Kal_B | 977K | 62 MB | 9.0% | 1.0% | 55.0% | 35.0% | 0.0% | 0.0% |
| Kal_C | 404K | 248 MB | 25.0% | 25.0% | 25.0% | 25.0% | 0.0% | 0.0% |
| Kal_D | 403K | 239 MB | 35.0% | 55.0% | 9.0% | 1.0% | 0.0% | 0.0% |
| Kal_E | 289K | 177 MB | 55.0% | 35.0% | 1.0% | 9.0% | 0.0% | 0.0% |
| KAL_D | 26,114K | 12.8 GB | 35.0% | 55.0% | 9.0% | 1.0% | 0.0% | 0.0% |

the use of MPI accelerates the building process in a substantial way. For example, when using 8 processors/executors, the improvement of the MPI version over the Spark version is bigger than a factor of 2×. When the the number of processes/executors is increased, the improvement of MPI with respect to Spark is reduced, although it is still better. For example, MPI reduces the execution time by 25% in the case of 64 processes. This reduction in the improvement is due to the limitations expressed by the Amdahl's law [40], as only one of the Meta-Cache building phases has been parallelized.

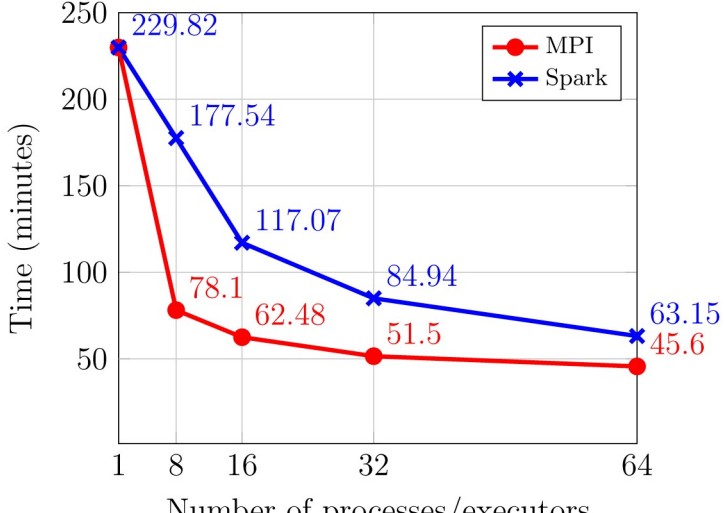

**Fig 3. Execution times of the MPI version and the Spark version to build the database AFS31RS90 with different number of processes.**

**Table 5. Peak memory (in GB per process/executor) for building database AFS31RS90.**

| Software Version | Number of processes/executors | | | | |
|---|---|---|---|---|---|
| | 1 | 8 | 16 | 32 | 64 |
| MetaCache-MPI | 135.00 | 56.91 | 49.65 | 33.66 | 31.84 |
| MetaCacheSpark | 135.00 | 175.12 | 100.07 | 68.94 | 45.14 |

On the other hand, results about memory consumption for the building phase for the selected data set are shown at Table 5. In this table, results with one process were obtained with the original version of MetaCache. We can see how the memory used by the MPI version is much smaller than the one used by Spark. For example, when using 8 cores, the memory used by the Spark version is more than 3× the amount of memory used by MPI, while the execution time is much smaller in the MPI version. As the number of cores increase, the difference in memory usage decreases. But even in the case of 64 cores, the difference is quite substantial, around 13 GB per process, which implies a total difference of **64 × 13 = 832** GB of total RAM consumption.

Regarding speed and memory consumption for the classification phase, both results can be observed at Table 6. Here, a comprehensive study regarding classification speed and RAM memory consumption is presented. Speed is shown in million reads classified per minute **(MR/m)** for different number of processes and threads used per process. Values for 32 processes/16 threads could not be obtained due to a limitation in the cluster queue system to run Spark. Regarding the RAM memory usage, as in the case of the building phase, it is presented as peak memory per process.

Taking this into account, it is also important to remember that the reference data used in this case is the database built from the AFS31RS90 data set and the KAL_D sausage sequences as input data to classification. As well as when building, results with one core are the ones obtained with the original MetaCache tool. Speed results show that the MPI version is much faster than the Spark version, between 2× and 3×, while the consumed memory for the MPI version is typically 50%—60% the amount of memory used in the Spark version.

Also, it is important to notice that the classification speed in the MPI version is higher than the one with the original MetaCache, being the opposite for MetaCacheSpark. This is because

**Table 6. Speed in MR/m and Peak memory (in GB per process) for querying database AFS31RS90 and dataset KAL_D in Big Data cluster.**

| MetaCache-MPI | | | | | | | | |
|---|---|---|---|---|---|---|---|---|
| | Speed in MR/m | | | | Peak memory | | | |
| Processes number | 1 | 8 | 16 | 32 | 1 | 8 | 16 | 32 |
| Threads per process | | | | | | | | |
| 4 | 2.00 | 5.39 | 5.77 | 5.26 | 116.71 GB | 40.00 GB | 36.50 GB | 32.20 GB |
| 8 | 3.89 | 7.80 | 7.91 | 7.31 | 116.82 GB | 40.92 GB | 36.13 GB | 32.60 GB |
| 16 | 6.90 | 8.18 | 8.75 | - | 117.12 GB | 41.50 GB | 36.58 GB | - |
| MetaCacheSpark | | | | | | | | |
| | Speed in MR/m | | | | Peak memory | | | |
| Executors number | 1 | 8 | 16 | 32 | 1 | 8 | 16 | 32 |
| Threads per executor | | | | | | | | |
| 4 | 2.00 | 1.35 | 1.63 | 1.54 | 116.71 GB | 110.87 GB | 64.68 GB | 48.20 GB |
| 8 | 3.89 | 2.63 | 2.44 | 2.35 | 116.82 GB | 111.71 GB | 65.02 GB | 50.80 GB |
| 16 | 6.90 | 3.11 | 2.94 | - | 117.12 GB | 111.62 GB | 65.72 GB | - |

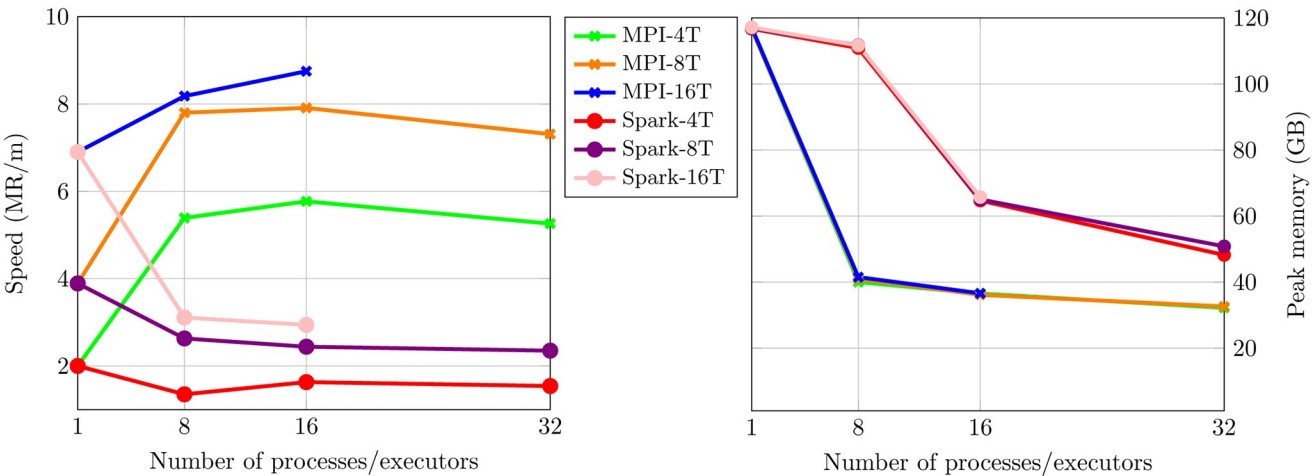

**Fig 4. Speed in MR/m (left) and Peak memory in GB per process (right) for querying database AFS31RS90 and dataset KAL_D in Big Data cluster.**

the piece of the database owned by each one of the processes is smaller than the one big hashmap owned by the unique process in MetaCache, thus it is faster to lookup in the smaller hashmaps. Also, in comparison to MetaCacheSpark, the small hashmaps owned by each processes are smaller in MPI (C++) than Java, with its consequent impact on performance. By using this approach, speeds very close to 9 MR/m were achieved, while consuming only 36.58 GB of peak RAM memory, for 16 processes and 16 threads per process. Meanwhile, Spark, with the same amount of processes/executors, is able to reach 2.94 MR/m while using 65.72 GB of peak RAM memory, which is almost twice the memory used by MPI. In order to show a more visual representation of these results Fig 4 is presented. In this Fig 4, to the left, results about the classification speed can be observed. To the right, results about memory consumption are shown. By looking at these plots two conclusions can be obtained. First, the difference regarding memory consumption when using 8 cores is overwhelming, almost 80 GB. The second, speed increases with the number of processes for the MPI version, except for 32 processes, probably because of the Amdahl's law. However, with Spark, speed decreases as the number of processes/executors grows.

## MPI scalability results

In this Section we present results about MetaCache-MPI scalability for both building and querying, using the HPC cluster FinisTerrae II at the Galicia Supercomputing Centre. Considering the Building phase for all data sets, the first computing time results are presented in Table 7. Here, execution times are divided in the time of adding the sequences into the database (which

**Table 7. Execution times (hh:mm:ss) when building databases in FinisTerrae II.**

|  | Adding sequences | | | | | Total | | | | |
|---|---|---|---|---|---|---|---|---|---|---|
| Proc. num. | 1 | 8 | 16 | 32 | 64 | 1 | 8 | 16 | 32 | 64 |
| Data set |  |  |  |  |  |  |  |  |  |  |
| AFS20 | 01:14:02 | 00:14:26 | 00:10:41 | 00:08:53 | 00:08:48 | 01:31:57 | 00:29:38 | 00:23:54 | 00:22:17 | 00:21:59 |
| AFS20RS90 | 02:46:51 | 00:31:41 | 00:28:02 | 00:26:41 | 00:25:52 | 03:09:42 | 00:48:51 | 00:43:54 | 00:41:43 | 00:40:31 |
| AFS31 | 02:07:34 | 00:23:52 | 00:20:35 | 00:17:24 | 00:15:07 | 02:31:17 | 00:41:47 | 00:38:26 | 00:34:12 | 00:31:37 |
| AFS31RS90 | - | 00:42:58 | 00:35:12 | 00:32:18 | 00:31:07 | - | 01:06:41 | 00:53:13 | 00:50:27 | 00:48:26 |

**Table 8. Consumed memory in GB when building for all databases in FinisTerrae II.**

| | Peak memory in GB | | | | |
|---|---|---|---|---|---|
| Number of processes | 1 | 8 | 16 | 32 | 64 |
| Data set | | | | | |
| AFS20 | 63.86 | 27.22 | 21.62 | 17.5 | 14.48 |
| AFS20RS90 | 110.04 | 43.96 | 44.54 | 30.59 | 24.55 |
| AFS31 | 90.62 | 39.64 | 35.76 | 28.24 | 24.13 |
| AFS31RS90 | 135.00 | 56.91 | 49.65 | 33.66 | 31.84 |

is step 2 from the "Parallel building phase" Section, i.e., the phase that has been actually parallelized) and the time involved in the whole building phase. The whole building time includes the time of adding sequences plus an extra time used for MPI communications, build the taxonomy, rank unranked targets and write the final database to disk. This extra time, that runs in a sequential way, is around 15-20 minutes, depending on the input data set.

In this Table, special focus should be put in the fact that all the databases can be created in less than one hour when using MPI. For example, when using the AFS20RS90 data set, the execution time of the original sequential versions is more than 3 hours, while in the MPI version this time decreases to 40 minutes using 64 cores. Although scalability is good with a small number of cores, when the number of cores increases, performance is devalued, mainly because of the impact of the I/O when adding the sequences. For example, with 8 processes, the speed-up just for adding sequences is between 5× and 5.4× for all the data sets. Regarding the total time with the same number of processes, speed up is around 3.5×. This value is lower than the speed-up of adding sequences again because of the Amdahl's law.

Results with one process could not be obtained for the AFS31RS90 data set, as the maximum available memory per node is 128 GB, and the process of building the database for this data set with the original MetaCache uses more than this amount of memory. However, it can be processed using various nodes in less than 50 minutes, which enables the possibility of building big databases that could not be created using the original MetaCache. We must highlight again that this is one of the big advantages of the MPI implementation.

Regarding the memory consumption for building, results are presented in Table 8. For the AFS31RS90 data set case with one process, the value is the indicated in Table 5. As it was stated before, results for the AFS31RS90 data set with MetaCache can not be obtained at the Finisterrae II, but the amount of memory is the same independently of the system used. Results show how the memory used to build database with dataset AFS20 decreases until barely 15 GB per process. With this amount of consumed memory, this database could be build in machines with 16 GB of RAM memory, which are very common nowadays.

This same approach can be taken for the AFS31RS90 data set. With this data set and 64 processes the peak memory is under 32 GB. Nowadays, typical memory in computing nodes for a cluster is between 64 and 128 GB or even more. As a final result, we can say that thanks to the distributed memory approach, databases that were very difficult to build in common nodes are now easily built in less time.

We now present the results regarding the classification speed and memory consumption. These results are shown in Table 9 where we query sequences from sausage KAL_D with all the databases. As the database for AFS31RS90 could not be built with MetaCache, results with one processor for this data set are not shown.

For the classification speed, again, as the access to the hashmap owned by each one of the MPI processes is faster than the access to the huge hashmap with just one process, speeds for

**Table 9. Speed in MR/m and Peak memory (in GB per process) for querying all databases and dataset KAL_D in FinisTerrae II.**

| AFS20 | | | | | | | | |
|---|---|---|---|---|---|---|---|---|
| | Speed in MR/m | | | | Peak memory | | | |
| **Processes number** | 1 | 8 | 16 | 32 | 1 | 8 | 16 | 32 |
| **Threads per process** | | | | | | | | |
| 1 | 0.98 | 3.19 | 3.72 | 4.05 | 46.99 GB | 31.66 GB | 28.27 GB | 28.40 GB |
| 4 | 3.39 | 9.09 | 9.50 | 9.41 | 47.15 GB | 31.40 GB | 27.97 GB | 27.66 GB |
| 8 | 6.48 | 11.26 | 11.82 | 10.65 | 47.38 GB | 30.70 GB | 28.83 GB | 28.27 GB |
| 16 | 11.2 | 14.68 | 13.05 | 12.63 | 47.81 GB | 32.05 GB | 29.37 GB | 28.51 GB |
| AFS20RS90 | | | | | | | | |
| | Speed in MR/m | | | | Peak memory | | | |
| **Process number** | 1 | 8 | 16 | 32 | 1 | 8 | 16 | 32 |
| **Threads per process** | | | | | | | | |
| 1 | 0.62 | 2.88 | 3.42 | 3.70 | 79.76 GB | 38.65 GB | 32.59 GB | 31.65 GB |
| 4 | 3.22 | 7.73 | 8.43 | 8.56 | 79.92 GB | 37.16 GB | 32.39 GB | 30.45 GB |
| 8 | 5.70 | 11.35 | 10.82 | 10.28 | 80.16 GB | 36.88 GB | 32.25 GB | 31.23 GB |
| 16 | 9.91 | 13.52 | 11.51 | 11.07 | 80.46 GB | 36.93 GB | 33.42 GB | 31.18 GB |
| AFS31 | | | | | | | | |
| | Speed in MR/m | | | | Peak memory | | | |
| **Process number** | 1 | 8 | 16 | 32 | 1 | 8 | 16 | 32 |
| **Threads per process** | | | | | | | | |
| 1 | 0.70 | 2.87 | 3.34 | 3.86 | 67.03 GB | 36.03 GB | 34.50 GB | 31.23 GB |
| 4 | 2.84 | 7.83 | 8.82 | 8.53 | 67.20 GB | 35.00 GB | 32.56 GB | 30.27 GB |
| 8 | 5.38 | 10.66 | 11.71 | 10.85 | 67.42 GB | 36.49 GB | 32.38 GB | 29.89 GB |
| 16 | 8.90 | 13.64 | 12.91 | 10.81 | 67.89 GB | 37.55 GB | 32.54 GB | 32.91 GB |
| AFS31RS90 | | | | | | | | |
| | Speed in MR/m | | | | Peak memory | | | |
| **Process number** | 1 | 8 | 16 | 32 | 1 | 8 | 16 | 32 |
| **Threads per process** | | | | | | | | |
| 1 | - | 2.67 | 3.08 | 3.49 | - | 39.87 GB | 37.59 GB | 32.73 GB |
| 4 | - | 6.90 | 7.61 | 8.00 | - | 40.0 GB | 36.50 GB | 32.20 GB |
| 8 | - | 10.06 | 10.32 | 9.04 | - | 40.92 GB | 36.13 GB | 32.60 GB |
| 16 | - | 12.97 | 11.21 | 10.54 | - | 41.50 GB | 36.58 GB | 33.05 GB |

MPI are higher than those for the original MetaCache. However, as the number of processes increase, speed from MetaCache gets closer, because, the more MPI processes, the more communications are needed, which implies a performance degradation. It is important to notice here that, in all the cases with 16 threads per process, a speed higher than 10 MR/m is achieved with the MPI version for all the databases, being close to 15 MR/m in the case of 8 processes and 16 threads per process for the AFS20 data set.

As for the memory consumption, there is relevant reduction of about half the usage in the MPI version, when using only 8 processes. For example, for the biggest database AFS31RS90, according to [25], the memory consumed for MetaCache during this phase is 117 GB, while the MPI version is able to decrease it to a peak memory of 41.50 GB using 8 cores and 16 threads per process. This indicates that, with these input data, MetaCache-MPI can run in typical computing clusters with a typical amount of memory per node (64-128 GB) and, at the same time, get classification speeds higher than the ones obtained with the original version of MetaCache.

**Table 10. Quantification results for the Klyo samples using the reference dataset AFS20.**

| Dataset | Classifier | Cattle | Pig | W.Buf. | Goat | Chicken | Turkey | Σ FP | Σ Dev |
|---------|-----------|--------|-----|--------|------|---------|--------|------|-------|
| KLyo_A | **Ground truth** | *14.0%* | 80.0% | 0.00% | 0.00% | 0.50% | 5.50% | | |
| | MetaCache | 16.6% | 71.5% | 0.04% | 0.02% | 0.60% | 4.64% | **0.28**% | **12.39**% |
| | MetaCache-MPI | 16.7% | 71.5% | 0.04% | 0.02% | 0.60% | 4.63% | 0.29% | 12.43% |
| | MetaCacheSpark | 16.9% | 71.2% | 0.04% | 0.02% | 0.60% | 4.64% | 0.32% | 12.99% |
| KLyo_B | **Ground truth** | 36.0% | 58.0% | 0.00% | 0.00% | 2.00% | 4.00% | | |
| | MetaCache | 37.6% | 51.0% | 0.12% | 0.04% | 2.05% | 2.99% | **0.50**% | **10.16**% |
| | MetaCache-MPI | 37.6% | 50.9% | 0.12% | 0.04% | 2.06% | 3.01% | 0.52% | 10.23% |
| | MetaCacheSpark | 37.9% | 50.5% | 0.12% | 0.04% | 2.06% | 3.02% | 0.60% | 11.11% |
| KLyo_C | **Ground truth** | 58.0% | 36.0% | 0.00% | 0.00% | 4.00% | 2.00% | | |
| | MetaCache | 57.7% | 27.1% | 0.16% | 0.06% | 3.56% | 1.16% | **0.95**% | **11.47**% |
| | MetaCache-MPI | 57.7% | 27.0% | 0.16% | 0.06% | 3.59% | 1.17% | 0.97% | 11.52% |
| | MetaCacheSpark | 57.7% | 26.9% | 0.16% | 0.06% | 3.63% | 1.18% | 0.95% | 11.48% |
| KLyo_D | **Ground truth** | 80.0% | 14.0% | 0.00% | 0.00% | 5.50% | 0.50% | | |
| | MetaCache | 74.7% | 10.9% | 0.23% | 0.08% | 4.66% | 0.33% | **0.93**% | **10.27**% |
| | MetaCache-MPI | 74.7% | 10.9% | 0.23% | 0.08% | 4.69% | 0.33% | 0.94% | 10.29% |
| | MetaCacheSpark | 74.7% | 10.8% | 0.23% | 0.08% | 4.69% | 0.33% | 1.09% | 10.58% |
| Average | MetaCache | | | **0.14**% | **0.05**% | | | **0.67**% | **11.07**% |
| | MetaCache-MPI | | | **0.14**% | **0.05**% | | | 0.69% | 11.12% |
| | MetaCacheSpark | | | **0.14**% | **0.05**% | | | 0.74% | 11.54% |

W.Buf: Water Buffalo, Σ FP: Sum of all false positive read classifications, Σ Dev: Sum of absolute deviations to the given meat composition (best results for each dataset in bold).

## Quantification results

In this Section, we present results about accuracy and quantification. To do this, we perform a comparison between the original MetaCache tool, MetaCacheSpark and MetaCache-MPI for all the input sausages with database AFS20. Results for MetaMache and MetaCacheSpark are the same from [25].

Results for KLyo A-D sausages can be observed at Table 10. We can see how results for MetaCache-MPI are very similar to the other versions. The results for the original Meta-Cache version are closer to the expected data, but the results obtained with the MPI version are almost the same, having very small differences. These differences are caused by ties among possible candidates when performing the classification. As we can see, results obtained with the MPI version are almost equivalent to the sequential version while, at the same time, the memory consumption and execution time are smaller using MetaCache-MPI.

Results for Kal A-E and KAL_D are shown at Table 11. Again, as in the previous case, results are very similar for all versions, but in this case MetaCacheSpark performs slightly worse, while MetaCache and MetaCache-MPI get results very close to the expected. For the KLyo case, it seemed that MetaCache is able to get slightly better results, but, in the Kal case, MetaCache-MPI seems to present a better behaviour. Again, we must highlight that reducing the execution times and memory consumption is essential for this kind of problems. Because of that, and as the results observed for all the input sausages are very similar, is very clear that the use of the MetaCache-MPI version has a lot of advantages over the sequential and the Spark counterparts.

**Table 11. Quantification results for the Kal samples using the reference dataset AFS20.**

| Dataset | Classifier | Cattle | Sheep | Pig | Horse | W.Buf. | Goat | Σ FP | Σ Dev |
|---|---|---|---|---|---|---|---|---|---|
| Kal_A | **Ground truth** | 1.00% | 9.0% | 35.0% | 55.0% | 0.00% | 0.00% | | |
| | MetaCache | 1.25% | 11.0% | 30.5% | 54.1% | 0.01% | 0.29% | **0.42**% | **8.13**% |
| | MetaCache-MPI | 1.25% | 11.1% | 30.5% | 54.1% | 0.01% | 0.29% | **0.42**% | 8.24% |
| | MetaCacheSpark | 1.27% | 11.1% | 30.3% | 54.1% | 0.01% | 0.29% | 0.45% | 8.42% |
| Kal_B | **Ground truth** | 9.0% | 1.00% | 55.0% | 35.0% | 0.00% | 0.00% | | |
| | MetaCache | 10.5% | 1.42% | 49.3% | 35.6% | 0.03% | 0.06% | **0.27**% | 8.43% |
| | MetaCache-MPI | 10.5% | 1.42% | 49.3% | 35.5% | 0.03% | 0.06% | **0.27**% | **8.39**% |
| | MetaCacheSpark | 10.6% | 1.42% | 49.1% | 35.7% | 0.03% | 0.06% | 0.30% | 8.92% |
| Kal_C | **Ground truth** | 25.0% | 25.0% | 25.0% | 25.0% | 0.00% | 0.00% | | |
| | MetaCache | 23.3% | 29.6% | 19.2% | 23.0% | 0.06% | 0.73% | **1.08**% | **15.28**% |
| | MetaCache-MPI | 23.3% | 29.7% | 19.2% | 22.9% | 0.06% | 0.73% | **1.08**% | 15.32% |
| | MetaCacheSpark | 23.5% | 29.6% | 19.0% | 22.9% | 0.06% | 0.73% | 1.18% | 15.32% |
| Kal_D | **Ground truth** | 35.0% | 55.0% | 9.00% | 1.00% | 0.00% | 0.00% | | |
| | MetaCache | 32.9% | 51.5% | 7.14% | 1.14% | 0.09% | 1.50% | **2.07**% | 9.62% |
| | MetaCache-MPI | 32.9% | 51.5% | 7.12% | 1.13% | 0.09% | 1.50% | **2.07**% | **9.61**% |
| | MetaCacheSpark | 33.2% | 51.2% | 7.03% | 1.13% | 0.09% | 1.49% | 2.23% | 9.91% |
| Kal_E | **Ground truth** | 55.0% | 35.0% | 1.00% | 9.00% | 0.00% | 0.00% | | |
| | MetaCache | 50.4% | 33.7% | 0.99% | 7.80% | 0.12% | 0.96% | **1.52**% | **8.55**% |
| | MetaCache-MPI | 50.5% | 33.7% | 0.99% | 7.79% | 0.12% | 0.96% | **1.52**% | **8.55**% |
| | MetaCacheSpark | 50.7% | 33.4% | 0.97% | 7.73% | 0.12% | 0.95% | 1.66% | 8.82% |
| KAL_D | **Ground truth** | 35.0% | 55.0% | 9.00% | 1.00% | 0.00% | 0.00% | | |
| | MetaCache | 30.3% | 49.6% | 7.27% | 1.16% | 0.08% | 1.25% | 1.38% | 13.36% |
| | MetaCache-MPI | 30.3% | 49.6% | 7.28% | 1.16% | 0.08% | 1.25% | **1.36**% | **13.34**% |
| | MetaCacheSpark | 30.4% | 49.5% | 7.25% | 1.16% | 0.08% | 1.26% | **1.36**% | 13.36% |
| Average | MetaCache | | | | | 0.07% | 0.80% | 1.12% | **10.56**% |
| | MetaCache-MPI | | | | | 0.07% | 0.80% | 1.12% | 10.57% |
| | MetaCacheSpark | | | | | 0.07% | 0.80% | 1.20% | 10.79% |

W.Buf: Water Buffalo, Σ FP: Sum of all false positive read classifications, Σ Dev: Sum of absolute deviations to the given meat composition (best results for each dataset in bold).

## Discussion

Food sequencing is becoming a very important area in metagenomics. Not only because of the quantification and identification of animal species in all kinds of animal-based food for quality control, but also because of the same procedures for other kinds of organisms, such as viruses, plants, fungi, or bacteria. This becomes extremely important, for example, in cases where avoiding the spread of a disease related with spoiled food becomes a global health problem. In these kind of situations, where speed and accuracy are a priority factor, the software introduced in this work can come into action and identify these cases in a very small amount of time, with an accuracy similar to MetaCache.

Furthermore, the required amount of species included in the study is also a very important handicap. This factor is directly related to the amount of memory used by the database. In the case of MetaCache-MPI, as it is a distributed memory approach, the consumed memory is distributed into several computing nodes, making it possible to handle data sets that otherwise would be impossible to tackle. All of this, without losing the accuracy that MetaCache can provide.

This accuracy has been tested throughout this work, where the authors have run a set of different experiments that have also been performed in [25]. In this way, results involving three different approaches were tested: one from MetaCache, the original sequential tool; other by using the Big Data engine Apache Spark; and the approach introduced in this work using MPI. The most important advantages of the MPI version are those related to memory consumption an the execution time (speed) for both database building and querying (classification).

These advantages can be observed in the results, where it is shown that the High Performance Computing approach by using MPI consumes less memory and time than the approach that uses the Spark Big Data engine. For example, for building the database with the biggest data set used in this work for 64 processes, MPI uses 70% of the RAM memory used by Spark, while spending only 72% of the time. Regarding classification/querying, MPI is faster by several millions of reads per minute. For example, for the biggest database and data set, using 32 processes and 8 threads per process, MPI is 3.11× faster, while using 64.17% of the RAM memory used by the Spark approach.

## Conclusions

In this work, the MetaCache-MPI tool has been introduced. It consists in a distributed memory approach to perform metagenomics analysis based on MetaCache, with the main advantages of a lower execution time and less memory consumption, and the main consequences of being able to create and query bigger databases that otherwise could not be created, with a significant reduction in time.

In order to test this implementation, experiments from [25] have been replicated and approached with the new MPI implementation. Results show how the new MPI version gets quantification and identification percentages similar to MetaCache, while being faster for both querying and building databases. This also applies to the comparison with MetaCacheSpark, where this work also establishes that MetaCache-MPI uses less RAM memory and is faster than the Apache Spark implementation. By relying on a MPI distributed memory approach, our software can scale efficiently towards large-scale collections of complex eukaryotic and bacterial reference genomes making this tool suitable for broad-scale metagenomic screening applications.

## Acknowledgments

The authors would like to thank Galicia Supercomputing Centre (CESGA) for the access to their supercomputing resources, and also CiTIUS systems administrators for their unconditional help.

## Author Contributions

**Conceptualization:** José M. Abuín.

**Formal analysis:** José M. Abuín.

**Funding acquisition:** José M. Abuín.

**Investigation:** José M. Abuín, Bertil Schmidt.

**Methodology:** José M. Abuín.

**Project administration:** José M. Abuín.

**Resources:** José M. Abuín.

**Software:** José M. Abuín.

**Supervision:** José M. Abuín, Nuno Lopes, Luís Ferreira, Tomás F. Pena, Bertil Schmidt.

**Validation:** José M. Abuín, Luís Ferreira, Bertil Schmidt.

**Visualization:** José M. Abuín, Nuno Lopes, Bertil Schmidt.

**Writing – original draft:** José M. Abuín, Nuno Lopes, Luís Ferreira, Tomás F. Pena, Bertil Schmidt.

**Writing – review & editing:** José M. Abuín, Nuno Lopes, Luís Ferreira, Tomás F. Pena, Bertil Schmidt.

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
