## [Decision Letter · Decision Letter 0]

20 Jul 2020

PONE-D-20-13398

Big Data in Metagenomics: Apache Spark vs MPI

PLOS ONE

Dear Dr. Abuín,

Thank you for submitting your manuscript to PLOS ONE. After careful consideration, we feel that it has merit but does not fully meet PLOS ONE’s publication criteria as it currently stands. Therefore, we invite you to submit a revised version of the manuscript that addresses the points raised during the review process.

We look forward to receiving your revised manuscript.

Kind regards,

Francisco Martínez-Álvarez

Academic Editor

PLOS ONE

Journal Requirements:

Reviewers' comments:

Reviewer's Responses to Questions

**Comments to the Author**

1. Is the manuscript technically sound, and do the data support the conclusions?

Reviewer #1: Yes

Reviewer #2: Yes

2. Has the statistical analysis been performed appropriately and rigorously? 

Reviewer #1: Yes

Reviewer #2: Yes

3. Have the authors made all data underlying the findings in their manuscript fully available?

Reviewer #1: Yes

Reviewer #2: Yes

4. Is the manuscript presented in an intelligible fashion and written in standard English?

Reviewer #1: Yes

Reviewer #2: Yes

5. Review Comments to the Author

Reviewer #1: I think that the authors deals with an important and interesting topic but the thing that they do is not really novel and many key paper in literature are not even mentioned (e.g. [1]) which does a very similar thing but with also the addition of OpenMP to MPI.

[1] Reyes-Ortiz, J. L. and Oneto, L. and Anguita, D., INNS International Conference on Big Data (INNS BIG DATA), Big Data Analytics in the Cloud: Spark on Hadoop vs MPI/OpenMP on Beowulf, 2015.

Reviewer #2: The authors use MPI to implement an existing algorithm and clearly demonstrate that the MPI implementation is more efficient than the original Spark version. The result is clearly applicable to the current food sample processing pipeline and also demonstrates the room for improvements for Spark applications in general.

6. PLOS authors have the option to publish the peer review history of their article (what does this mean?). If published, this will include your full peer review and any attached files.

Reviewer #1: No

Reviewer #2: **Yes: **Junhao Li

---

## [Author Response · Author response to Decision Letter 0]

26 Aug 2020

The text included here is the same than the one in the "Response for reviewers" document.

Reviewer #1 Comments:

I think that the authors deals with an important and interesting topic but the thing that they do is not really novel and many key paper in literature are not even mentioned (e.g. [1]) which does a very similar thing but with also the addition of OpenMP to MPI.

[1] Reyes-Ortiz, J. L. and Oneto, L. and Anguita, D., INNS International 

Conference on Big Data (INNS BIG DATA), 

Big Data Analytics in the Cloud: Spark on Hadoop vs MPI/OpenMP on Beowulf, 2015.

Response: We agree with the reviewer that there are more works that deal with similar problems and we have added the given reference. However, we were not able to find any similar comparison related to metagenomics or Animal Food Sequencing (AFS), so we could not add other references.

Reviewer #2 Comments:

The authors use MPI to implement an existing algorithm and clearly demonstrate that the MPI implementation is more efficient than the original Spark version. The result is clearly applicable to the current

food sample processing pipeline and also demonstrates the room for improvements for Spark applications in general.

Response: Indeed, as the reviewer says, there is room for improvements in general Spark applications related with genomics and other scientific problems. To show this is, in fact, one of the main objectives of this

paper.

---

## [Editor Report · Decision Letter 1]

14 Sep 2020

Big Data in Metagenomics: Apache Spark vs MPI

PONE-D-20-13398R1

Dear Dr. Abuín,

We’re pleased to inform you that your manuscript has been judged scientifically suitable for publication and will be formally accepted for publication once it meets all outstanding technical requirements.

Kind regards,

Francisco Martínez-Álvarez

Academic Editor

PLOS ONE

Additional Editor Comments (optional):

The authors have properly addressed all the concerns and the paper can now be accepted in its current form.
---

## [Editor Report · Acceptance letter]

25 Sep 2020

PONE-D-20-13398R1 

Big Data in Metagenomics: Apache Spark vs MPI 

Dear Dr. Abuín:

I'm pleased to inform you that your manuscript has been deemed suitable for publication in PLOS ONE. Congratulations! Your manuscript is now with our production department. 

Kind regards, 

on behalf of

Dr. Francisco Martínez-Álvarez 

Academic Editor

PLOS ONE